# Unveiling a Microexon Switch: Novel Regulation of the Activities of Sugar Assimilation and Plant-Cell-Wall-Degrading Xylanases and Cellulases by Xlr2 in *Trichoderma virens*

**DOI:** 10.3390/ijms25105172

**Published:** 2024-05-09

**Authors:** Cynthia Coccet Castañeda-Casasola, María Fernanda Nieto-Jacobo, Amanda Soares, Emir Alejandro Padilla-Padilla, Miguel Angel Anducho-Reyes, Chris Brown, Sereyboth Soth, Edgardo Ulises Esquivel-Naranjo, John Hampton, Artemio Mendoza-Mendoza

**Affiliations:** 1Faculty of Agriculture and Life Sciences, Lincoln University, Lincoln 7647, New Zealand; coccet987@hotmail.com (C.C.C.-C.); amaanda_soares@hotmail.com (A.S.); emiratos123@hotmail.com (E.A.P.-P.); sereyboth.soth@lincolnuni.ac.nz (S.S.); edgardo.esquivel@lincoln.ac.nz (E.U.E.-N.); john.hampton@lincoln.ac.nz (J.H.); 2Laboratorio de AgroBiotecnología, Universidad Politécnica de Pachuca, Carretera Pachuca-Cd. Sahagún, km 20, ExHacienda de Santa Bárbara, Zempoala 43830, Mexico; anducho@upp.edu.mx; 3Servicio Nacional de Sanidad, Inocuidad y Calidad Agroalimentaria, Centro Nacional de Referencia Fitosanitaria, Tecamac 55740, Mexico; 4Plant & Food Research, Lincoln, 74 Gerald Street, Lincoln 7608, New Zealand; fernanda.jacobo@plantandfood.co.nz; 5Department of Biochemistry, School of Biomedical Sciences, University of Otago, Dunedin 9054, New Zealand; chris.brown@otago.ac.nz; 6Centro de Ciencias Genómicas, Universidad Nacional Autónoma de México, Cuernavaca 04510, Mexico; 7Unit for Basic and Applied Microbiology, Faculty of Natural Sciences, Autonomous University of Queretaro, Queretaro 76230, Mexico

**Keywords:** alternative splicing, cellulase, microexons, plant symbiosis, solid-state fermentation (SSF), submerged fermentation (SmF), transcription factor, xylanase

## Abstract

Functional microexons have not previously been described in filamentous fungi. Here, we describe a novel mechanism of transcriptional regulation in *Trichoderma* requiring the inclusion of a microexon from the *Xlr2* gene. In low-glucose environments, a long mRNA including the microexon encodes a protein with a GAL4-like DNA-binding domain (Xlr2-α), whereas in high-glucose environments, a short mRNA that is produced encodes a protein lacking this DNA-binding domain (Xlr2-β). Interestingly, the protein isoforms differ in their impact on cellulase and xylanase activity. Deleting the *Xlr2* gene reduced both xylanase and cellulase activity and growth on different carbon sources, such as carboxymethylcellulose, xylan, glucose, and arabinose. The overexpression of either *Xlr2-α* or *Xlr2-β* in *T. virens* showed that the short isoform (Xlr2-β) caused higher xylanase activity than the wild types or the long isoform (Xlr2-α). Conversely, cellulase activity did not increase when overexpressing *Xlr2-β* but was increased with the overexpression of *Xlr2-α*. This is the first report of a novel transcriptional regulation mechanism of plant-cell-wall-degrading enzyme activity in *T. virens.* This involves the differential expression of a microexon from a gene encoding a transcriptional regulator.

## 1. Introduction

Fungi, as versatile organisms, employ sophisticated mechanisms to thrive in diverse environments, with their adaptability intricately tied to the precise regulation of gene expression [1,2]. A vital aspect of this regulatory tapestry is the controlled expression of plant-cell-wall-degrading enzymes (PCWDEs), indispensable for the breakdown of complex polysaccharides in plant cell walls [3,4]. The precise coordination of PCWDE expression ensures fungal fitness and their ability to efficiently utilise plant biomass as a nutrient source. Moreover, this coordination is crucial during plant colonization for various plant-associated organisms, including pathogens and symbionts [4].

At the epicentre of the regulatory network governing PCWDE expression is a set of transcription factors, acting as molecular conductors that finely tune the expression of PCWDE-encoding genes [4,5,6]. Transcriptional regulators of PCWDEs have been extensively studied due to their biotechnological implications in enzymatic production [7,8,9]. Various transcriptional regulators of PCWDEs have been identified and characterised in fungi, including *Neurospora crassa* [10], *Aspergillus* spp. [7,11], and *Trichoderma* spp. [6,12]. In *T. reesei*, the regulation of genes encoding for PCWDEs involves the coordination of at least five transcriptional activators—Xyr1, Ace2, Ace3, BgIR, and the Hap2/3/5 complex—and two repressors, Ace1 and the carbon catabolite repressor Cre1 [4,13].

Microexons are relatively new players in the regulation of gene expression [14]. These are short exons (usually ≤36 base pairs), shown to occur in large numbers in some eukaryotic genomes [15,16]. Formerly perceived as inconspicuous entities, microexons have now gained recognition as influential contributors to the intricate orchestration of gene expression. They play a crucial role in diverse biological systems, including blood-dwelling schistosome worms [15], human tissues, the nematode *Caenorhabditis elegans*, the fruit fly *Drosophila melanogaster*, and the mustard plant *Arabidopsis thaliana* [17]. Recently, a conserved microexon program that regulates glucose homeostasis in pancreatic cells was identified [18,19]. However, the characterisation and identification of microexons in fungi are relatively limited [20,21,22,23,24,25] and have only been experimentally tested in *Schizosaccharomyces pombe* [20,22].

Alternative splicing (AS) is a mechanism that can generate a repertoire of mRNA and protein isoforms from a single gene [26]. Even though only about 6% of fungal genes exhibit AS [27], this mechanism plays a crucial role in cell identity and differentiation in eukaryotes, particularly in plants and animals [28,29]. In the context of microexons, their alternative splicing is linked to the regulation of protein interaction networks in developing neurons, and the misregulation of microexons has been implicated in neurological disorders such as autism [30,31]. Additionally, AS can be modulated by the environment and by transcriptional and epigenetic mechanisms [26]. Multiple transcripts encoding different protein isoforms can also be generated by alternative transcription start sites.

*Trichoderma* spp. are cosmopolitan fungi distributed in almost all ecosystems and are one of the most successful fungal biofertilisers and biological control agents used in agriculture [32,33,34]. The role of PCWDEs and their associated regulatory control mechanisms in *Trichoderma* have only recently begun to be explored. Initial transcriptomics [35,36] and proteomics studies [37,38] have investigated the molecular communication mechanisms between *Trichoderma* and its environment. For example, during the *Trichoderma* plant interaction, fungal hemicellulases, represented by cellulases and xylanases, have been identified as the most upregulated and abundantly produced molecules [34,35,36,37,39].

In the current study, we investigated and characterised a novel and conserved Zn(II)_2_Cys_6_ transcription activating factor from *T. virens*, which we refer to herein as Xlr2 (from transcription factor similar to XlnR, xylanase regulator 1, [7]). We also investigated the role of two predicted isoforms of the Xlr2 protein and their regulation via the expression of a “switch-like” microexon through a glucose-dependent mechanism in *T. virens*.

This study reveals a significant finding: a microexon, once considered on the genomic periphery, plays a crucial role in regulating Xlr2, a novel transcription factor in *T. virens*, adding a nuanced layer of control to the expression of PCWDEs. Moreover, this paper provides a facet of genetic variation for future breakthroughs in biotechnology and deepening the understanding of fungal biology through revealing the mysteries of microexon-mediated regulation. In addition, our results refine the current knowledge related to molecular mechanisms in fungi, with a strong focus on the control of PCWDEs, thereby suggesting new approaches for agronomy and bioenergy.

## 2. Results

### 2.1. Xlr2 Is a Novel Zn(II)_2_Cys_6_ Transcription Factor Conserved in Filamentous Fungi

Zinc cluster transcriptional regulator proteins belong to one of the largest families of transcription factors (TFs) in fungi [40]. The Zn(II)_2_Cys_6_ transcription factors are zinc finger proteins that contain a DNA-binding domain with six cysteine residues bound to two zinc atoms [41]. The TF XlnR/Xlr1/Xyr1 possesses the Zn(II)_2_Cys_6_ DNA-binding domain found in diverse fungal TFs [42]. This domain is an important positive activator present in many TFs that regulates the expression of PCWDEs [43]. In this study, we characterised a novel and conserved Zn(II)_2_Cys_6_ transcription factor from *T. virens* (ID:47927, https://genome.jgi.doe.gov/pages/search-for-genes.jsf?organism=TriviGv29_8_2 (accessed on 25 January 2022)), which we have named xylanase regulator 2 (Xlr2) due to its similarity to XlrR. In the Joint Genome Institute (JGI) database, the annotated polypeptide from the Xlr2-encoding gene comprised 575 amino acid residues, with a classical fungal transcription factor and a putative transmembrane domain. However, the N-terminal region corresponding to the Zn(II)_2_Cys_6_ (also called Gal4) domain, which is required for binding to the DNA and is present in different transcription factors from the same family in fungi, was not annotated. We mapped the reads from our RNA-seq data, generated from different conditions in *T. virens* Gv29.8 [44], to the reference genome in the JGI database (https://genome.jgi.doe.gov/TriviGv29_8_2/TriviGv29_8_2.home.html (accessed on 25 January 2022)) and manually annotated the *Xlr2* gene and transcripts (Figure 1A). Our manual annotation based on the transcriptome data showed that the transcript produced under low glucose contained two additional exons. The corrected full-length Xlr2 translation product (875 amino acids) includes a Zn(II)_2_Cys_6_ DNA-binding domain that is annotated as a xylanase regulator in diverse fungi in the NCBI database (e.g., *T. reesei* Figure 1B). Xlr2 contains a putative transmembrane domain, the N-terminal Zn(II)_2_Cys_6_ activating domain (Figure 1B, ZnFD), which is common to many eukaryotic transcription regulators [45], and a fungal transcription factor domain (fungal TF, PFAM, PF04082, Figure 1B). Except for the transmembrane domain, these domains are also present in the xylanase regulator Xyr1/Xlnr from *T. reesei*, as well in the corresponding ortholog in *T. virens* (JGI Protein ID: 58714) (Figure 1B). The Zn(II)_2_Cys_6_ binuclear cluster from Xlr2 is conserved, and a comparison of Xlr2 with other Zn(II)_2_Cys_6_ proteins, like AraR, GlnR and XlnR, showed conserved amino acids additional to the six conserved cysteines (yellow) common to Zn(II)_2_Cys_6_ proteins (Figure 1C).

Xlr2 homologs are present in other ascomycetes, including pathogens of insects, mites, and nematodes such as *Hirsutella* spp. *Cordyceps brongniartii* (anamorph: *Beauveria brongniartii*), *Ophiocordyceps sinensis*, *Metarhizium anisopliae*, and *Moelleriella* spp.; plant pathogens like *Magnaporthe oryzae*, *Verticillium longisporum*, *Fusarium oxysporum*, and *Fusarium verticillioides*; and cellulolytic fungi like *Pseudogymnoascus* spp., which are living as saprotrophs. However, Xlr2 and its orthologs (red) are distinct from the xylanase regulator Xyr1 from *T. reesei* (or *T. virens*) and its orthologs (blue), which are in a different branch in the phylogenic tree, and Xlr2-like and Xyr1/Xlnr-like proteins are more closely related to each other than the GalR/AraR regulators (green and purple) (Figure 1D).

### 2.2. Xlr2 Is Involved in Morphology and Sugar Assimilation in T. virens

To analyse the role of the *Xlr2* (PP712105) gene in *T. virens* Gv29.8, the open reading frame (ORF) was deleted by substitution with the hygromycin phosphotransferase gene under the control of the TrpC promoter. The substitution was confirmed using a Southern blot analysis (Appendix A) and internal primers to amplify the open reading frame, which verified the absence of the *Xlr2* gene. Three independent strains with identical phenotypes were selected. These strains did not contain the open reading frame and had single integrations of the hygromycin resistance gene. The *Xlr2* gene deleted strain (Δxlr2) was selected, as it exhibited differences in growth and sporulation behaviour in minimal medium containing glucose compared to the parental strain. Additionally, microscopic and morphological differences were observed. Notably, the Δxlr2 strain showed a significant reduction in pellet size, and the typical swelling structures observed in the wild type (hereinafter referred to as WT) were absent (Appendix A). To assess the contribution of Xlr2 in sugar assimilation, *T. virens* WT and Δxlr2 strains were grown on Vogel’s medium supplemented with different carbon sources. When grown on arabinose, sucrose, lactose, glycerol, carboxymethylcellulose (CMC), or birch xylan as the carbon source, Δxlr2 showed a significant reduction in growth compared to the WT (Figure 2A). However, significant differences in growth between the WT and Δxlr2 were not observed when galactose, xylose, maltose, or sorbitol was used as the carbon source (Figure 2B). These data suggest that Xlr2 might be involved in the assimilation or metabolism of pentose and hexoses but not in the galactose oxidation–reduction pathway.

### 2.3. Xlr2 Transcription Factor Partially Regulates PCWDE Activity in T. virens

To assess the role of Xlr2 in the regulation of PCWDEs, cellulase and xylanase enzymatic activity was determined for 72 h under submerged fermentation (SmF) with either glucose, CMC, or xylan as the carbon source (Figure 3). No xylanase or cellulase activities were observed when glucose was used in either the WT or Δxlr2 strain. The most effective inducers for *T. virens* xylanase and cellulase production peaked at different times. Xylan induced the highest xylanolytic activity of 60.87 ± 0.43 AU/mL for the WT and 34.87 ± 0.99 AU/mL for Δxlr2 (Figure 3A). In contrast, when using CMC as the carbon source, the cellulase activity was 9.24 ± 0.25 AU/mL in the WT and 6.3 ± 1.29 AU/mL in Δxlr2 (Figure 3B).

### 2.4. A Microexon in Xlr2 Is Conserved in the Trichoderma Genus

RNA-sequencing (RNA-seq) reads were mapped to the *T. virens* Gv29.8 genome to facilitate the interpretation of expressed transcripts, the identification of alternative splices for genes, and the discovery of novel transcripts. By comparing different RNA-seq datasets from *T. virens* Gv29.8 growth under different environmental conditions, including Vogel’s minimal medium with or without glucose as a carbon source, we observed that the transcript structure of *Xlr2* is regulated by glucose repression (Figure 4A). Specifically, the cultivation of *T. virens* in a glucose-deprived medium induces an mRNA including the first exon-encoding part of the DNA-binding domain and a second 20-nucleotide microexon encoding another part of it (5′ TCCTTGTCGTCGTTGTGAAG-3′) (Figure 4A,B). In the presence of glucose, this microexon is not expressed, and the first exon and third to fifth exons are expressed as two transcripts. There is no evidence of reads linking exons 1 and 3 in this experiment, nor in other experiments where glucose was present (Figure 4A). As we previously noted, this microexon was not predicted in the annotated JGI consensus gene model for *Xlr2* (Figure 1A), probably because its exceptionally small size will elude most de novo gene prediction software.

To investigate the conservation of this microexon in orthologs of *Xlr2* in other *Trichoderma* species, we searched for its presence in the publicly available *Trichoderma* genomes (https://genome.jgi.doe.gov/programs/fungi/index.jsf; https://www.ncbi.nlm.nih.gov/bioproject/PRJNA314460 (accessed on 30 April 2024)). The annotation of the coding sequence (CDS) of Xlr2 orthologs from *T. guizhouense* (OPB37586.1) shows the presence of the spliced microexon (Appendix A); however, the Xlr2 orthologs in eight additional *Trichoderma* species, (*T. atroviride* (Triat2_229644), *T. reesei* (TRQM6a_60282), *T. harzianum* CBS 226.95 (Triha1_79686), *T. asperellum* CBS 433.97 (Trias1_137241), *T. asperellum* TR356 (Triasp1_375582), *T. citrinoviride* TUCIM 6016 (Trici4_1109904), *T. harzianum* TR274 (Trihar1_701679), and *T. longibrachiatum* ATCC 18648 (Trilo3_1437635), are smaller than predicted, compared to our reannotation of Xlr2 from *T. virens*. The N-terminal Zn(II)_2_Cys_6_ domain was not annotated, nor was the microexon. Although *T. gamsii* T6085 (Trigam1_8287) was reported to contain the Zn(II)_2_Cys_6_ domain, the microexon and the bordering introns were not annotated in this ortholog of Xlr2. As a first step towards identifying this domain in the Xlr2 orthologous genes, we analysed 5 kb sequence data from the 5′-flanking regions upstream of the reported transcriptional start site.

Alignment of the cDNA obtained from our RNA-seq data (cDNA *Xlr2*), the predicted mRNA from *T. guizhouense* (cDNA OPB3758) and *T. parareesei* (OTA07803.1), and the extended 5′ region of the *Trichoderma* orthologous genes showed the presence of the microexon sequence in all *Xlr2* orthologs from the *Trichodema* species analysed here (Appendix A). This microexon is highly conserved among *Trichoderma* species, encoding PCRRCE, including two of the cysteines (C) in the Zn(II)_2_Cys_6_ domain (Appendix A). The substitution pattern across the *Trichoderma* species indicates a protein-coding exon, with third-position codon changes in most positions (Figure 4B). Moreover, the introns bordering the microexon in all the orthologous sequences have the canonical donor–acceptor 5′ G/A|GTR and AG|T 3′ reported in other fungi [49] (Figure 4C). Using RNA-seq data from *T. reesei* (JGI *Trichoderma reesei* QM6a ID: 60282), there is evidence that intron 1 is retained in some transcripts, depending on the sugar source (Appendix A). This may indicate that the microexon is inefficiently spliced.

To identify the correct translation product from longer transcripts in all the other *Trichoderma* species, we manually removed the introns identified, including those bordering the microexon. We confirmed that orthologs of *Xlr2*-encoding genes contain the Zn(II)_2_Cys_6_ finger domain, the fungal transcription factor domain, and the transmembrane domain (Appendix A).

### 2.5. Differential Expression of the Microexon in Xlr2 Represents a Novel Regulatory Mechanism in Sugar Assimilation and Cell-Wall-Degrading Enzymes in Trichoderma

The differential expression of the *Xlr2* microexon generates two possible Xlr2 isoforms. The expression of the microexon in *Xlr2* (e.g., in the absence of glucose) generates an ORF of 2628 bases, called here long *Xlr2*. The translation product of the long *Xlr2* is a polypeptide of 895 amino acid residues with a Zn(II)_2_Cys_6_ finger domain in the N-terminal region and the other domains mentioned above. In contrast, two transcripts appear to be made when the microexon is not expressed (e.g., in the presence of glucose) (Figure 5). One of exon 1 would encode part of the DNA-binding domain and terminate at the unspliced junction in all *Trichoderma* (TG/GTGA). The transcript for exons 3–6 could encode a shorter ORF that starts 717 bases downstream of the start codon from the long *Xlr2*. However, this has multiple out-of-frame upstream AUG codons, so it may not efficiently produce a protein. This generates an ORF of 2082 bases with a translational product comprised of a 693-amino-acid polypeptide, which does not contain the Zn(II)_2_Cys_6_ finger-binding domain but which does contain the fungal transcription factor domain and the transmembrane domain (Figure 5).

We confirmed the expression of the microexon *Xlr2* by RT-PCR using cDNA generated from mycelium grown in Vogel’s minimal medium, with or without glucose. The primer combination AM-LU650 and RL-LU51LR (indicated as primers “a” and “b” in Figure 5A and the sequence in Appendix A) allowed microexon expression to be identified, as it bound to the 5′ region of the gene body from the long *Xlr2* transcript, which is missing in the short *Xlr2* transcript. As observed in Figure 5B (Appendix A), a strong signal was obtained in the medium without glucose (VM-C), while only a weak signal was shown in Vogel’s minimal medium supplemented with glucose (VMC). To assess the role of these two versions of a transcript, both versions of the *Xlr2* were independently overexpressed in *T. virens* using the constitutive promoter tef1 from *T. virens*. Several transformants were obtained, but only one of each version was chosen for further studies. These strains were called overexpressing (OE) strain OEXlr2-α-L2A (long *Xlr2* transcript) and OEXlr2-β-S23A (short *Xlr2* transcript). qRT-PCR was used to analyse the expression levels of these two variants, and we observed that both overexpressing strains showed higher levels of *Xlr2* expression when compared to the WT (Figure 5B).

First, the growth phenotypes for the OEXlr2-α-L2A and OEXlr2-β-S23A strains were compared to the WT strain using different carbon sources. A significant increase in growth of the OEXlr2-α-L2A strain over that of the WT strain was observed when xylose, maltose, sorbitol, CMC, and birch xylan were used as carbon sources. Moreover, the OEXlr2-α-L2A strain produced darker green spores and more abundant aerial growth than the WT in xylose, maltose, and sorbitol (Figure 6A). The OEXlr2-β-S23A strain showed a significant increase in growth when xylose, CMC, and xylan were used as carbon sources. Using CMC and birch xylan also significantly increased the growth of the OEXlr2-α-L2A and OEXlr2-β-S23A strains (Figure 6B).

### 2.6. OEXlr2 Regulates Cellulase and Xylanase Biosynthesis in Trichoderma virens under Solid-State and Submerged Fermentation

To examine the influence of the *Xlr2* gene on the regulation of the production of xylanolytic and cellulolytic enzymes of *T. virens*, the WT and OE strains were subjected to solid-state fermentation (SSF) and submerged fermentation (SmF) assays. The development of the strains was evaluated under both conditions, monitoring the xylanolytic and cellulolytic activity in birch xylan and CMC, respectively. During the SmF, xylanase activity in the WT strain was detected, with the highest activity (45.49 ± 1.52 AU/mL) reported at 72 h, while the maximum cellulase activity (9.24 ± 0.25 AU/mL) was reported at 96 h under the test conditions (Figure 7A). The OEXlr2-α-L2A strain had higher xylanase (59.43 ± 1.32 AU/mL) and cellulase activity (12.78 ± 1.94 AU/mL) than the WT strain; however, the highest xylanase activity overall was observed in the OEXlr2-β-S23A strain (102.60 ± 0.98 AU/mL), indicating that overexpression of the *Xlr2* short gene increased xylanase activity compared to the WT strain. The OEXlr2-β-S23A strain exhibited a higher growth in CMC than the WT strain, demonstrating the capacity of this strain to grow in this complex carbon source due to its ability to produce the necessary enzymes (cellulases) for the assimilation of this carbon source (Figure 7B).

WT and OE strains were grown in the SSF using a corncob as support. The xylanase and cellulase activities are shown in Figure 7B. The OEXlr2-α-L2A strain showed the highest xylanase activity (756.07 ± 44.99 AU/gdm), which was observed after 148 h of culture, and was 1.4 times greater activity than the WT strain (562.32 ± 21.61 AU/gdm). The OEXlr2-β-S23A strain showed the maximum xylanolytic activity at 120 h (335.01 ± 27.29 AU/gdm), while the highest reported cellulase activity was 64.22 ± 9.97 AU/gdm, which was higher than that of the WT strain (59.87 ± 1.88 AU/gdm).

## 3. Discussion

This study presents the first report of a “switch-like” microexon in fungi and its role in regulating lytic and enzyme activity in the beneficial fungi *T. virens*. Specifically, we found that the microexon distinguishes two distinct isoforms of a novel transcription factor (Xlr2), referred to herein as short *Xlr2-β* and long *Xlr2-α*. We also found that the overexpression of each isoform results in the differential regulation of xylanase and cellulase activity in this species. Finally, we report the conservation of this microexon, at least in the genus *Trichoderma*.

### 3.1. Identification and Characterization of a Novel and Conserved Zn(II)_2_-C_6_ Transcription Activating Factor from T. virens

The identification of the gene *Xlr2* in *T. virens*, belonging to the Zn(II)_2_Cys_6_ transcription factor family, presents an intriguing avenue for understanding regulatory mechanisms in fungi. This family, characterised by a DNA-binding domain featuring six cysteine residues coordinated with two zinc atoms, plays a crucial role in transcriptional activation [50]. Notably, the transcription activator XlnR/Xlr1/Xyr1, encompassing the Zn(II)_2_Cys_6_ DNA-binding domain, serves as a pivotal regulator of (hemi) cellulolytic genes in various fungi [42].

In our study, we characterised Xlr2 as a novel and conserved Zn(II)_2_Cys_6_ transcription activating factor in *T*. *virens* (Protein ID:47927). This characterization was based on its sequence similarity to XlnR and its homologs identified through NCBI-BLAST, which are described as “putative transcription activators similar to XlnR”. Given XlnR’s established role in regulating PCWDEs, including in *T. reesei*, the identification of a similar factor in *T. virens* suggests a potential regulatory role in xylan and cellulose degradation pathways.

It is noteworthy that while the annotated *Xlr2* gene comprises a predicted polypeptide of 575 amino acid residues with classical transcription factor and transmembrane domains, the N-terminal region corresponding to the Zn(II)_2_Cys_6_ domain, characteristic of PCWDE regulators, was initially overlooked. However, through meticulous analysis, including RNA-seq mapping to the *T. virens* genome, we identified the presence of this crucial N-terminal region and highlighted the absence of its annotation.

This discovery underscores the importance of employing complementary approaches, such as RNA-seq mapping, to validate and refine genome annotations. Furthermore, the presence of the Zn(II)_2_Cys_6_ domain in Xlr2 suggests its potential involvement in regulating the expression of PCWDEs in *T. virens*, expanding our understanding of fungal cellulolytic mechanisms.

In conclusion, the characterization of Xlr2 as a novel Zn(II)_2_Cys_6_ transcription-activating factor in *T*. *virens* provides valuable insights into the regulatory network governing cellulose degradation in fungi. Future investigations aimed at elucidating the precise mechanisms by which Xlr2 modulates gene expression will deepen our understanding of fungal biology and may inform strategies for improving biofuel production and plant biomass degradation.

### 3.2. The Transcription Activator Xlr2 Is Conserved in Trichoderma

Like Xyr1, GalR, AraR, and Gal4p, Xlr2 belongs to the Zn(II)_2_Cys_6_ family. These transcription factors regulate multiple processes, including development and metabolism [51]. In this study, Xlr2 presents a DNA-binding domain called a binuclear Zn(II)_2_Cys_6_ cluster, which is common to many eukaryotic transcription regulators [45], as well as the domain of a fungal-specific transcription factor and a transmembrane domain. By comparing orthologous genes in other *Trichoderma* species, we identified the presence of the same domains in the encoding proteins from these orthologs (Appendix A).

Xlr2 homologs are present in other ascomycetes, including fungal pathogens of plants and animals, as well as in several species of typical cellulolytic fungi, including those distantly related. However, Xlr2 is phylogenetically separated from Xlr1 orthologs of *T. virens* (JGI ID: 58714) and Xyr1 orthologs of *T. reesei* (JGI ID: 1222084), which are found in a different branch in the phylogenetic tree (Figure 1D). XlnR and AraR are more similar, controlling the genes in response to D-xylose and L-arabinose, respectively, [52,53]. Both regulators are involved in the regulation of genes encoding (hemi)cellulases, as well as enzymes of the pentose catabolic and pentose phosphate pathways [54]. The ancestral importance of hydrolytic enzymes for *Trichoderma* species may be responsible for this grouping due to the different lifestyles of *Trichoderma* species as either mycoparasites or as plant cell wall degraders [32]. It has also recently been suggested that some of these enzymes in *Trichoderma* may have originated in plants [39].

### 3.3. Xlr2 Is Involved in the Regulation of Xylanase and Cellulase Activity

PCWDEs are one of the most abundant molecules during *T. virens* plant–root interactions, suggesting that they play an important role during colonisation [35,36,37]. These PCWDEs include glucanases, β-glucosidases, proteases, and endoglucanases, as well as xylanases and cellulases [34,39], and transcription factors tightly regulate their production.

The regulation of the expression of cellulases and xylanases in *T. reesei* has been widely investigated, with the process controlled by a set of transcription factors that include positive regulators [11,12,55,56,57,58]. Xlr1 is an important regulator in multiple fungal systems and controls the catabolism of D-xylose and the xylanolytic system [59]. However, there are some variations to this; for example, the ortholog XlnR in *M. oryzae* (Xlr1) does not regulate its xylanolytic system and seems to be only involved in D-xylose catabolism [60]. Meanwhile, in *N. crassa*, XlnR is not essential for the use of cellulose and only modulates the expression of some cellulolytic genes [10]. In *T. reesei*, Xyr1, a homolog of XlnR from *A. niger*, regulates the expression of the (hemi)cellulolytic genes [11]. However, further knowledge regarding transcription factors operating in other species, such *T. virens*, has not yet been reported.

Our results suggest that Xlr2 is involved in the regulation of xylanases and cellulases, since the deletion of the corresponding gene (*Xlr2*) decreased the growth of *T. virens* on these polysaccharides and it had reduced enzymatic activity with respect to the WT. Specifically, we found that the deletion mutant of *Xlr2* from *T. virens* was affected in its ability to grow effectively in xylan but not in D-xylose. This observation of defective growth on xylan was supported by an observed decrease in xylanase activity in the Δxlr2 deletion mutant (Figure 3A). This suggested that Xlr2 has a role in xylan catabolism but not D-xylose catabolism and perhaps suggests that Xlr2 in *T. virens* has a similar role to Xlr1 in *P. canescens* [61].

Involvement of the xylanolytic activators, XlnR (*T. reesei*) and Xyr1 (*A. niger*), has also been reported in the regulation of the expression of cellulolytic and hemicellulolytic genes [51]. Several studies show that Xyr1 is the primary positive transcriptional activator of cellulase and that the deletion of Xyr1 not only suppresses cellulase activity but also xylanase activity and the formation of β-mannanases [12,43]. In our study, the reduction in growth of the Δxlr2 strain compared to the WT in media containing either xylan or carboxymethylcellulose as the carbon source suggested that Xlr2 is required for the assimilation of both polysaccharides. This reduction correlated with the reduction in xylanase and cellulase activities in the Δxlr2 strain (Figure 3), thereby indicating that Xlr2 not only controls the pentose phosphate pathway but might also regulate encoding genes for PCWDEs involved in the oxido-reductive D-galactose catabolic pathway and regulated by another transcription factor.

### 3.4. The Regulation of PCWDEs via Fungal Perceptions of Plant Cell Wall Polysaccharides

In fungi, the regulation of PCWDEs is controlled via a mechanism involving the fungal perception of derivatives of the PCW sources that are used by the coloniser [4,53]. We found that the genetic deletion of *Xlr2* in *T. virens* impeded the growth of the fungus on media containing xylan or CMC, which suggested an essential role for this gene in the assimilation, processing, or transport of these polysaccharides (i.e., that this gene is not involved in the catabolic process of xylose but instead in the processing/transport or perception of xylan). Thus, it seems that Xlr2 is not involved in regulating any of the enzymes involved in xylose metabolism via the pentose catabolic pathway. As a corollary to this, however, it has been shown that in *A. nidulans*, the expression level of the xylose reductase (XyrA) gene is not reduced in the absence of XlnR and is highly expressed in the absence of AraR [53]. It is presumed that, in this instance, the concentration of the carbon source intervenes in the production of xylanases for the assimilation of hemicellulolytic compounds. This considers that the accumulation of high concentrations of D-xylose generates a relatively slow metabolic process, turning the accumulation into a feedback mechanism [58], whereby the high D-xylose concentrations have a repressing effect [57,62].

Lactose, a soluble carbon source that promotes the expression of cellulolytic genes in *T. reesei* [4], is cleaved by β-galactosidases into glucose and galactose. It is most likely that Xlr2 is involved in the assimilation of glucose but not galactose, which demonstrates that the *Xlr2* gene does not participate in the galactose oxidation–reduction pathway or that the regulation of specific genes involved in galactose catabolism is controlled by transcription factors other than Xlr2 (i.e., GalR or AraR) [52,63].

Growth of the strain Δxlr2 was also impeded on arabinose, showing a reduction in radial growth, thereby indicating that Xlr2 is somehow involved in the pentose phosphate pathway. The deletion of the XlnR gene from *A. nidulans* demonstrated that XlnR intervenes in the regulation of LarA (L-arabinose reductase), LadA (L-arabitol dehydrogenase), and LxrA genes (L-xylulose reductase) [53], which are involved in the catabolism of arabinose, suggesting that Xlr2 may have a similar function. However, further work needs to be conducted to test this hypothesis.

### 3.5. Role of the Long and Short Isoforms of Xlr2 and the "Switch-like" Microexon

We found that in the case of the *Xlr2* gene, the microexon acted as a switch to be included in a "long" isoform (Xlr2-α), which differed in function from the "short" isoform (Xlr2-β) in the presence or absence of glucose. Microexons are a class of very short-length (3–30 base pairs) exons [16,28,30]. They have been predicted in three fungal genes and shown to function in *S. pombe*. They are usually multiples of three nucleotides in length, which enables both inclusion and exclusion to maintain the reading frame [64]; they are often associated with the production of alternative protein isoforms [28]. Microexons also regulate protein–protein interactions [35,65] by altering their structure, stability, or subcellular location [30]. Interestingly, the misregulation of alternative splicing by a microexon in CPEB4, a key protein that coordinates the expression of hundreds of genes required for neuronal activity, has been identified in the brains of individuals with autism [31]. This observation indicates that a correct balance of protein isoforms within the cells is important for the precise physiology of the cells. In our study, the expression of the microexon in *Xlr2*, which controls Xlr2-α or Xlr2-β protein isoforms, was found in *T. virens* under different culture conditions, i.e., with or without glucose (Figure 5). The truncation at the N-terminus of the Xlr2-β protein eliminated the Gal4 regulatory domain, albeit with the shorter protein still retaining sequence similarity with the XlnR regulators, as well as the domain of a fungal-specific transcription factor (PFAM, PF04082). This suggests that the short version may be involved in a separate regulatory mechanism or form a protein with a different function. In distinct contrast, the Xlr2-α protein contains a zinc cluster DNA-binding domain. Zinc clusters can interact with DNA as monomers or homodimers/heterodimers [50], with this class of transcription factor reported to regulate different cellular processes required for the survival of the microorganism, including the metabolism of sugars and amino acids, respiration, the cell cycle, chromatin remodelling, and stress response [66,67,68,69].

### 3.6. Carbohydrate Assimilation in the Long and Short Xlr2 Isoforms

The *T. virens* long and short Xlr2 isoform mutants differed in their growth in different carbon sources (Figure 7). For instance, the OEXlr2-α-L2A strain showed greater growth on xylose than the WT strain. Growth aberrations on pure oligosaccharides, such as D-xylose, have been demonstrated previously in *T. reesei*, where they induce the expression of cellulolytic and hemicellulolytic genes (i.e., *bgl1* [4], *bgl2* and *bxl1* [70], *cbh1* [71], and *xynI* [72]), while at higher concentrations they act as a repressor through CreA/Cre1 [11]. The induction of (hemi)cellulolytic genes by oligosaccharides, such as arabinose (*abf1*, *abf2*, and *abf3*) [64], lactose (*gal1*, *cbh1*, and *cel5b*) [73], and galactose (*agl1*, *agl2*, and *ax1*) [4], has also been reported. Although L-arabinose is catabolised by filamentous fungi through the pentose catabolic pathway [70], and D-galactose is catabolised through the oxidoreductive pathway [71] and/or the Leloir pathway [74,75], our results did not show the participation of Xlr2 in the catabolism of either of these carbon sources (i.e., the radial growth of both of the *T. virens* strains overexpressing the short or long isoforms was comparable to the WT strain). The regulator AraR/Ara1 controls the catabolism of L-arabinose, but in *Aspergillus* the total control of the use of pentoses is shared with the cellulolytic regulator (hemi) XlnR [54]. This is not the case in *Pyricularia oryzae*, where ARA1 completely controls the catabolism of L-arabinose and its release of plant biomass, apparently without any requirement for Xlr1 [76]. However, differences in the regulation of their use have been reported in *A. niger*; L-arabinose was regulated by AraR, while D-xylose was regulated by XlnR, although overlaps occur between the regulation networks of AraR and XlnR [52,63]. Meanwhile, in *T. reesei*, it was shown that Xyr1 was not involved in the response to D-galactose.

On the other hand, the induction of cellulolytic and hemicellulolytic enzymes by Xyr1 is restricted, especially by lactose in *T. reesei* [11,77], where the expression of the genes involved in its assimilation are induced. Thus, if D-lactose is the cellulase inducer, the growth of OEXlr2-α-L2A and OEXlr2-β-S23A is limited, probably due to low extracellular β-galactosidase activity. There are differences in metabolic preferences among *Trichoderma* spp. For instance, unlike the observations in *T. reesei*, lactose did not improve the production of cellulases in *T. harzianum* [78]. Thus, the catabolism of individual oligosaccharides in various *Trichoderma* spp. appears to be strongly influenced by the type of regulator, fungal species, and growth conditions, and perhaps future research will reveal further involvement of microexons.

### 3.7. Overexpression of the Long and Short Xlr2 Isoforms Affects Catabolism

The OE strains (OEXlr2-α-L2A and OEXlr2-β-S23A) also showed significantly more growth than the WT strain in the medium with CMC and birch xylan, which means that the overexpressing strains can increase the degradation of the polymer. The presence of cellulose, xylan, or mixtures of biopolymers present in the culture medium has been previously reported to cause abundant production of cellulolytic and xylanolytic activities by *T. reesei* [79,80]. These differences in growth indicate that both the Xlr2 long and short isoforms regulate the cellulase and xylanase genes of *T. virens* due to their ability to assimilate these carbon sources. The birch xylan used during this study consists mainly of xylose and arabinose, indicating that hemicellulolytic enzymes (β-xylanases, β-xylosidases, and L-arabinofuranosidases) are being produced. In contrast, the CMC indicated the production of cellulases (e.g., glucanases and glucosidases) by the OE strains in order to assimilate those polysaccharides. These results point strongly to the fact that Xlr2 acts as a general regulator of the cellulolytic and xylanolytic enzymatic system of *T. virens*. In addition to the carbon source, other environmental and physiological factors affect the enzymatic production of the fungus, such as Xyr1, which is responsible for regulating the expression of cellulases and hemicellulases [80].

### 3.8. The Xlr2 Microexon Regulates Cellulase and Xylase Activity during SmF and SSF

Marked differences were apparent in the fermentation profiles of the two strains of *T. virens*, overexpressing either the long or short Xlr2 isoform. The OEXlr2-α-L2A strain was the best producer of xylanase and cellulase enzymes, with xylanase activity significantly higher than that observed in the WT strain (Figure 7). Interestingly, overexpression of the Xlr2 short isoform in *T. virens* in the OEXlr2-β-S23A strain only increased xylanase activity during the SmF. Conversely, in the SSF, only cellulase activity increased with respect to the WT strain. While there are reports of overexpression studies involving fungal TFs [81,82,83,84], no other comparative studies are reported between SmF and SSF for the overexpression of a transcription factor. Thus, this study is the first report to compare the effects of these two different fermentations on the overexpression of a transcription factor.

### 3.9. Cellulase and Xylanase Activity Is Affected by the Fermentation Technique

In this study, the activities of xylanase and cellulase in *T. virens* cultivated in SmF and SSF were compared to understand if SSF improves the production of enzymes. We noted differences in the enzyme profiles produced using the two fermentation procedures (SSF and SmF), with the SSF generally yielding much higher enzyme activity than the SmF. It is reported that SSF generally allows a higher production of crude enzymes compared to SmF [85,86]. An essential biological factor in favour of SSF is the use of high concentrations of the carbon source without affecting microbial growth and catabolic repression [87], which seems to limit the production of enzymes in SmF [88]. The use of agro-industrial waste in SSF allows several advantages focused on its management, energy consumption, and the production of metabolites compared to SmF [89]. Using birch xylan in the culture medium during the SmF allowed the induction of xylanases. In contrast, the use of a corncob in the SSF, which provided a carbon source due to its high content of hemicelluloses [90], made it a very attractive support for obtaining enzymes, as it also promoted the induction of the genes involved in xylanase expression. Increased enzymatic activities have been detected in SSF with the use of agro-industrial waste by microorganisms of the genus *Trichoderma*, and enzymes such as xylanases [91,92,93,94,95,96], cellulases [97,98,99,100], lipases [101], polygalacturonase [102], and pectinase have been considered in comparative studies between SSF and SmF.

In the *T. virens* strains overexpressing the long or short Xlr2 isoforms, the OEXlr2-α-L2A strain showed higher enzymatic activities in the SSF compared to the SmF (Figure 7). Overexpression studies using Xyr1 in *T. harzianum* to induce enzymatic activity have also demonstrated the influence of this positive regulator in the production of cellulolytic enzymes. When sugarcane bagasse was used as a carbon source, the overexpressed Xyr1 gene generated higher expression levels and enzymatic productivity when both strains (WT and OE) were cultivated under SmF [100]. Similarly, overexpression of Mtxyr1 (homolog Xyr1) in *Myceliophthora thermophila* [78] increased the xylanolytic activity in the presence of glucose and a corncob with respect to the WT strain, albeit with no change in the levels of endoglucanase activity [82]. The TF CXRD (cellulolytic and xylanolytic regulator) resulted in a 49.3 to 2230% enhanced production of cellulase and xylanase, except for 75% less xylanase at 2 days, compared with the *Penicillium oxalicum* parental strain when cultured on solid medium containing wheat bran plus rice straw [101].

The ability to grow in a wide range of carbon sources allows significant variability in enzyme production since the gene expression and secretion of enzymes are directly dependent on the diverse substrates. This finding will contribute to understanding the molecular mechanism of the regulation of xylanases and cellulases of *T. virens* under SSF.

## 4. Materials and Methods

### 4.1. Fungal Strains

*Trichoderma virens* Gv29.8 strain (kindly provided by Prof. Charles Kenerley, Texas A&M University, College Station, TX, USA), *T. virens* Xlr2 (PP712105), Δxlr2, and Xlr2 overexpression strains (OEXlr2-α and OEXlr2-β strains, Microbiology Lab, Lincoln University, Canterbury, New Zealand) were maintained at −80 °C in glycerol 25%, and spores were propagated on Potato Dextrose Agar (PDA) at 25 °C. The *Escherichia coli* Top10F strain was used for all cloning purposes and propagated at 37 °C on Luria–Bertani (LB) supplemented with the corresponding antibiotic. Transformants were selected with 100 μg/mL hygromycin B (Hyg) or 700 μg/mL Geneticin (Gibco, Thermo Fisher Scientific, Waltham, MA, USA).

### 4.2. Growth Conditions

Conidia of *T. virens* Gv29.8 WT, OE-Xlr2-α-L2A, and OE-Xlr2-β-S23A overexpression for inoculating liquid cultures were collected from mycelia grown for 7 d at 28 °C on PDA. Unless otherwise stated, Vogel’s medium (VM) (adapted from Vogel, [103]) was used for the solid and liquid minimal medium using 2% of the sugar indicated in the experiments.

For gene expression experiments, strains were first grown in 500 mL flasks containing 100 mL of Vogel’s medium supplemented with 2% glucose at 150 rpm and 28 °C. After 48 h of culture, mycelia from each strain were filtered and washed, and equal amounts of each strain were suspended in a VM containing 2% glucose. Samples were immediately taken after the transfer (t = 0).

To determine hyphal growth on agar plates, agar plates containing Vogel’s medium supplemented with 2% of different sugars (glucose, galactose, xylose, arabinose, maltose, sucrose, lactose, sorbitol, glycerol, carboxymethyl cellulose, or xylan) were inoculated in the centre with 1 × 10^6^ spores of *T. virens* WT or mutants. Plates were incubated at 25 °C, and colony diameter was measured at 24, 48, and 72 h. The experiments were repeated three times for consistency and reliability.

Kinetics of xylanase and cellulase activity in the SmF were carried out in a mineral medium modified by Téllez-Téllez et al. [104]. For the SSF, a corncob was used as the agro-industrial support. Conidia (1 × 10^6^) of OExlr2long-L2A and OExlr2-S23A strains were inoculated in 100 mL of medium in 500 mL flasks and incubated on a rotary shaker at 150 rpm at 28 °C.

Vogel medium. This medium contained (per litre) 30 g of Na_3_C_6_H_5_O_7_, 50 g of KH_2_PO_4_, 20 g of NH_4_NO_3_, 2 g of MgSO_4_ 7H_2_O, and 1 g of CaCl_2_ 2H_2_O. Then, 1 mL of a trace element solution containing the following compounds was added per litre of medium: citric acid (50 g/L), ZnSO_4_·7H_2_O (50 g/L), Fe(NH_4_)_2_(SO_4_)_2_·7H_2_O (10 g/L), CuSO_4_·5H_2_O (2.5 g/L), MnSO_4_·H_2_O (0.5 g/L), H_3_BO_3_ (0.5 g/L) NaMoO_4_·2H_2_O (0.5 g/L), and 1 mL of Biotin Solution (0.1 mg).

Mineral medium modified (g/L). This medium contained KH_2_PO_4_, 0.6; MgSO_4_·7H_2_O, 0.5; K_2_HPO_4_, 0.4; FeSO_4_·7H_2_O, 0.05; MnSO_4_·H_2_O, 0.05; and ZnSO_4_·7H_2_O, supplemented with yeast extract (5.0 g/L) and birch xylan (5.0 g/L) or CMC (5.0 g/L).

### 4.3. Construction of Plasmids

pAMT-30 contains the phosphotransferase of hygromycin under the control of the TrpC promoter. This plasmid was used to create the following vectors [37].

pGEMT-47927LF-HY vector. A 1.1 kb region of the 5′ region of 47927 (Gene Identification number in the JGI *T. virens* Gv29.8 V2 genome) available in the JGI gene was amplified using *T. virens* Gv29.8 genomic DNA as the template and the primer combination 47927-LFR-SfiI and 47927-OPL (Appendix A). The resulting PCR product was digested with an SfiI restriction enzyme and ligated to the 1.4 kb SfiI hygromycin resistance cassette fragment from pAMT-30 (Mendoza-Mendoza, unpublished). To generate the 47927LF-HY product, the ligation product was amplified with Phusion Taq polymerase (Thermo Fisher Scientific, MA, USA) using the primer combination 47927 LF L and oAM-LU347 (Appendix A). The resulting PCR product was gel purified, A-tailed, and ligated into plasmid pGEMT, generating plasmid pGEMT-47927LF-HY. The plasmid was amplified in *E. coli* Top10F’ and sequenced. pGEMT-47927LF-HY was used for the overlap transformation of *T. virens* Gv29.8 alongside pGEMT-47927RF-YG.

pGEMT-47927RF-YG vector. A 1.1 kb region of the 3′ region of the 47927 gene was amplified from *T. virens* Gv29.8 genomic DNA using primers the 47927 RF L SfiI and 47927 OPR. The vector was created similarly to pGEMT-47927LF-HY, except the primer combination 47927 RF-R and oAM-LU348 was used. Plasmid pGEMT-47927RF-YG was used for overlap transformation alongside pGEMT-47927LF-HY.

The OEXlr2-α (long version) and OEXlr2-β (short version) vectors were used to create the overexpressing vectors. The TrichoGate system was used as a Golden Gate subcloning strategy adapted for *Trichoderma* and other filamentous fungi [105]. Different TrichoGate vectors were used, including (1) pTrichoGate-3, containing the constitutive promoter from the translation elongation factor 1 alpha (tef1α); (2) pTrichoGate-16, containing the terminator of the nopaline synthase (T-nos); and (3) pTrichoGate-20, containing the neomycin phosphotransferase II (nptII), which confers resistance to the antibiotic Geneticin. These were amplified with Phusion Taq polymerase using the primers indicated in Appendix A. The corresponding PCR products were subcloned in the EcoRV restriction site from a mutated version of pUC19, where the BsaI restriction site from the ampicillin resistance gene was previously mutated (kindly provided by Keen Sohn). These components (tef1α, T-nos, and nptII) were designed with bordering BsaI recognition sites with distinct and specific cleaving outside sites for the BsaI restriction enzyme but allowed for directional subcloning with any desired gene [105]. These plasmids were called pUC19-tef1α, pUC19-T-nos, and pUC19-nptII. The recipient plasmid was pAGM1311, provided by Addgene, which contains a kanamycin resistance gene [106].

The Xlr2-α and Xlr2-β fragments were amplified from genomic DNA. A single digestion–ligation step was conducted using purified PCR products from Phusion^®^ High-Fidelity DNA Polymerase (Thermo Fisher Scientific, MA, USA). PCR products of Xlr2-α and Xlr2-β ORFs were analysed in agarose gel, with the fragments purified using the Wizard^®^ SV Gel and PCR Clean-Up System (Promega, Auckland, New Zealand) to extract DNA before using 100 ng of individual PCR products (Xlr2-α or Xlr2-β), which were combined with pUC19-tef1α, pUC19-T-nos, pUC19-nptII, and pAGM1311 and incubated in a thermocycler for 5 h at 37 °C with 10 U T4-ligase (10 U) and BsaI (10 U) in a T4 ligase buffer, followed by 5 min at 50 °C and 5 min at 80 °C, with a final stage of 4 °C. The Xlr2-α ORF was amplified by PCR using the primers AM-LU702 and AM-LU704, and the Xlr2-β ORF was obtained using the primer combination AM-LU703 and AM-LU704. The ligation restriction was transformed into *E. coli* Top10F and spread in LB supplemented with kanamycin, X-Gal, and IPTG. White colonies were selected, and the plasmids were isolated using the Macherey Nagel Miniprep Kit (Dueren, Germany), following the manufacturer’s instructions.

### 4.4. Protoplast Preparation and Transformation

Three PDA plates were covered with cellophane discs, and 100 μL of spores of the WT strain were inoculated per plate and incubated at 25 °C for 14–24 h. The discs containing the mycelium were suspended in 10 mL of an enzyme solution (0.24 g of cellulase and 0.5 g of Glucanex^®^ (Sigma-Aldrich, Darmstadt, Germany) in 50 mL of an osmotic solution: 50 mM CaCl, 0.5 M to 0.7 M mannitol, 50 mM MES, pH 5.5). The digestion mixture was incubated for 4 h at 25 °C with shaking at 150 rpm. Subsequently, the protoplasts were purified by filtration and centrifugation (10 min at 6000× *g*). After suspending the protoplasts in 300 μL of the osmotic solution, they were again centrifuged and finally suspended to give a concentration of 1 × 10^8^ protoplasts/mL.

Protoplast transformation of the WT strain was carried out by inoculating 130 μL of protoplasts (around 1 × 10^7^ protoplasts) with 10 μg of DNA (plasmid linearised with the XbaI enzyme in each of the constructions). It was incubated on ice for 20 min, and then 130 μL of the polyethylene glycol (PEG) solution (40% PEG 4000 in 0.7 M osmotic mannitol) was added for 30 min at room temperature. The transformation reaction was added to 20 mL of regeneration medium (PDA with 0.7% agar, 0.5 M sucrose) supplemented with 700 μg/mL of Geneticin, then incubated at 25 °C for 7 d. Colonies were selected following purification in a selective medium (Geneticin), ensuring the isolation of a single stable transforming colony.

### 4.5. Gene Expression Studies

Total RNA extraction from *T. virens* (WT and overexpressing strains) was obtained by using the RNeasy Plant Mini Kit (Qiagen, Hilden, Germany) according to the manufacturer’s instructions. The DNA was removed using the TURBO DNA-free™ Kit (Ambion, Thermo Fisher Scientific, MA, USA). The concentration of RNA was measured using a Nanodrop 2000 spectrophotometer (Thermo Fisher Scientific, MA, USA). The synthesis of cDNA from total RNA was performed using the Super Script III First-Strand synthesis system for the RT-PCR Kit (Invitrogen™, Thermo Fisher Scientific, MA, USA), according to the manufacturer’s instructions.

For quantitative real-time PCR, the primer combinations AM-LU703 and AM-LU708 for the Xlr2 gene and oCC7-UPP and oCC8-UPP (Appendix A) for the 18S gene were amplified in a StepOne Real-Time PCR System (Applied Biosystems, Waltham, MA, USA) using PowerUp SYBR Green Master Mix (Applied Biosystems, MA, USA). All PCR reactions were performed in triplicate, in a total volume of 10 μL for 30 cycles, under the following conditions: denaturation, 95 °C, 15 s; annealing, 60 °C, 1 min; extension, 72 °C, 1 min; and final extension, 72 °C, 3 min. Three biological samples were used for each strain. Threshold cycles (CT) were determined using the System Software (Applied Biosystems StepOne and StepOnePlus Real-Time PCR Systems Software v2.2.2), and CT values were calculated using the 18S gene as a housekeeping control. Data were expressed as 2(−∆∆CT) [40]. Nine values per sample were used for statistical analysis.

### 4.6. Xylanase and Cellulase Activity in SSF and SmF

To measure the enzymatic activity in the SmF, 40 mL of culture medium (see above for medium used) was inoculated with 1 × 10^6^ spores/mL of the corresponding strain in 125 mL Erlenmeyer flasks. The culture was incubated at 28 °C on an orbital shaker at 150 rpm. The samples were collected in triplicate every 24 h for 5 d using a Neubauer Counting Chamber (Sigma-Aldrich, Darmstadt, Germany).

For the SSF, a corncob, 16 mesh size, was used as the substrate. Before use, the corncob was washed three times with hot bi-distilled water, followed by two washes with cold distilled water, and dried at room temperature for 24 h followed by incubation at 50 °C for 48 h. Five grams of the substrate was placed in a 125 mL flask with a 75% humidity. The substrate was inoculated with 1 × 10^6^ spores/g of dry substrate. All flasks were incubated at 28 °C for 5 d in triplicate.

### 4.7. Obtaining Enzymatic Extracts

The enzymatic extract of both types of fermentation was obtained every 24 h. In the case of the SmF, the extract was obtained by filtration. From the filtered liquid, 3 mL was recovered in 1.5 mL Eppendorf tubes and stored at 4 °C. In the case of the SSF, the substrate was washed with 30 mL of distilled water and shaken vigorously for 10 min. Finally, 3 mL of the enzyme extract was taken and stored at 4 °C.

### 4.8. Analytical Techniques

The xylanolytic and cellulolytic activity in the enzymatic extracts was determined by the dinitrosalicylic acid method [107]. The xylanase and cellulase activities were determined by incubating the enzyme extract at 50 °C and using 0.5% (*w*/*v*) birch xylan (Sigma-Aldrich, Darmstadt, Germany) and 0.5% (*w*/*v*) carboxymethylcellulose (Sigma-Aldrich, Darmstadt, Germany) as substrates in 100 mM acetate buffer (pH 5.3). A unit (U) of xylanase/cellulase was defined as the amount of enzyme required to release 1 μmol of xylose equivalent/glucose per minute under the conditions tested.

### 4.9. RNA-Seq Analysis of the Alternative Splicing

RNA-seq data were aligned to the *T. virens* Gv29.8 v2 reference genome using Bowtie2 [108] and Tophat 2 [109]. The resultant BAM files were sorted, indexed, and loaded into the IGV Browser for visualization and interpretation [48]. Using IGV, we inspected regions of interest, examined read alignments, and compared sequencing data across multiple samples and conditions.

### 4.10. Statistical Analysis

The treatments were arranged using a completely randomised experimental design. All data were subjected to a simple analysis of variance (ANOVA). The means were compared using the Fisher’s least significant difference (LSD) method (*p* ≤ 0.05), with the statistical package STATGRAPHICS Centurión XV.II used to identify whether or not there were significant differences between the results obtained for each evaluated strain. Each biological and experimental test was performed as the mean value of three repetitions. The LSD method [110] was used to compare statistically significant differences (*p* < 0.05).

## 5. Conclusions

This study presents the first report of a previously unrecognised layer of regulation in the Zn(II)_2_Cys_6_-type transcription factor Xlr2 from *T. virens*. Initially annotated without the characteristic Zn(II)_2_Cys_6_ DNA-binding domain, this transcription factor was shown to possess a microexon that plays a crucial role in its regulatory mechanism. The findings significantly contribute to understanding the regulatory network governing sugar assimilation and plant-cell-wall-degrading enzyme (PCWDE) activities in filamentous fungi. We identified and characterised a novel gene, Xlr2, in *T. virens*, belonging to the Zn(II)_2_Cys_6_ transcription factor family. This family is known for its role in regulating the expression of PCWDEs in fungi. Xlr2 is conserved in *Trichoderma* species and other ascomycetes, including fungal pathogens and cellulolytic fungi. Our findings demonstrated that Xlr2 is involved in the regulation of xylanases and cellulases, key components of PCWDEs. The deletion of Xlr2 results in decreased growth of *T. virens* on xylan, accompanied by a reduction in xylanase and cellulase activities. The study also suggested that Xlr2 plays a role in the assimilation of xylan and cellulose, highlighting its importance in the fungal response to plant cell wall polysaccharides.

Furthermore, we examined the regulatory role of Xlr2 in different carbon sources. Xlr2 is implicated in the assimilation of glucose, but not galactose, and is also involved in the pentose phosphate pathway. Additionally, Xlr2 responded to specific carbon sources and is not directly linked to the catabolism of all tested sugars. A noteworthy finding is the identification of long and short isoforms of Xlr2, namely Xlr2-α and Xlr2-β, generated through alternative splicing involving a microexon. These isoforms exhibit different functions in the presence or absence of glucose. The results suggested that the microexon acts as a switch-like microexon [30], influencing the inclusion of the long or short isoform based on environmental conditions. Furthermore, the overexpression of both long and short isoforms of Xlr2 resulted in increased growth on certain carbon sources and the enhanced production of cellulases and xylanases. The study also examined the effects of overexpression in SmF and SSF, revealing that SSF generally leads to higher enzyme activity. This finding highlighted the importance of the fermentation technique in optimising enzyme production. In summary, this research provides valuable insights into the regulatory role of Xlr2 in *T. virens*, shedding light on its involvement in the regulation of PCWDEs, carbon source utilisation, and the influence of alternative splicing on its function. The findings contribute to the understanding of the molecular mechanisms underlying fungal–plant interactions and have implications for the industrial production of enzymes for various applications.

## Figures and Tables

**Figure 1 ijms-25-05172-f001:**
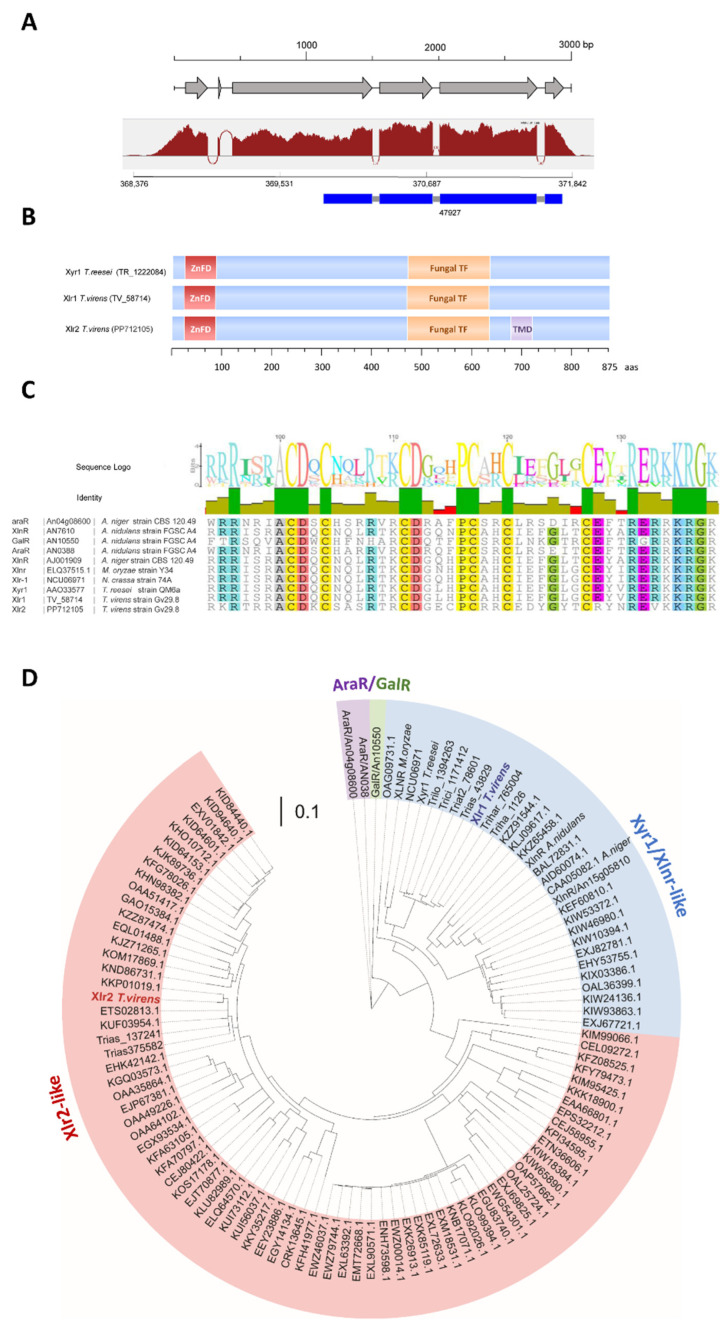
Structure and phylogenetic relationship of the transcription factor Xlr2 from the *Trichoderma virens* Gv29.8. Xlr2 (PP712105) gene of *T. virens*. (**A**) The solid blue boxes indicate the protein prediction by JGI for the Protein ID 47927 from *T. virens* Gv29.8. The Sashimi plot generated was based on RNA-seq data obtained from samples of *T. virens* Gv29.8 grown in Vogel’s minimal medium without glucose for 3 h and is indicated in red. The exons are represented by grey arrows. (**B**) Schematic representation of Xlr2 from *T. virens* and the Xyr1 transcription factor from *T. reesei* and its orthologous in *T. virens*: N-terminal Zn(II)_2_Cys_6_ activating domain (ZnFD), fungal transcription factor domain (fungal TF) and transmembrane domain (TMD). (**C**) Alignment of the DNA-binding domain Gal4 (Zn(II)_2_Cys_6_) (from Xlr2 and the corresponding other transcription factors involved in cell-wall-degrading enzymes and sugar metabolism). The percentage of sequence identity given for each conserved domain is more than 80% of the sequences in the alignment; a coloured background indicates conserved amino acids. (**D**) Phylogenetic analysis of the Xlr2 and related homologs of *T. virens* and other organisms. MUSCLE was used to generate alignments for phylogenies, and Geneious Tree Builder was used to produce distance trees using the Neighbour-Joining method. The Resampling method was set to Bootstrap with 1000 replicates. The phylogenetic tree was then annotated and edited using Interactive Tree of Life (iTOL) version 6. The scale bar refers to a phylogenetic distance of 0.1 nucleotide substitutions per site [46,47].

**Figure 2 ijms-25-05172-f002:**
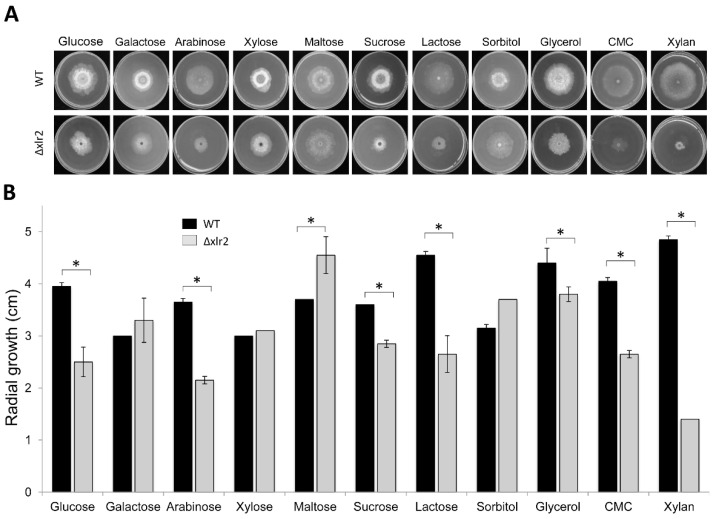
The radial growth profile of the wild-type (WT) and deleted (Δxlr2) strain of *Trichoderma virens* Gv29.8 grown on different carbon sources. (**A**) Radial growth of WT and Δxlr2 strains in Petri dishes after 3 days at 25 °C. (**B**) Statistical analysis of the radial growth of WT and Δxlr2 strains. The results shown represent the average value of three repetitions, and the error bars indicate the standard error of the mean. * Indicates that the Δxlr2 strain had a significantly slower radial growth than the WT (*p* < 0.05).

**Figure 3 ijms-25-05172-f003:**
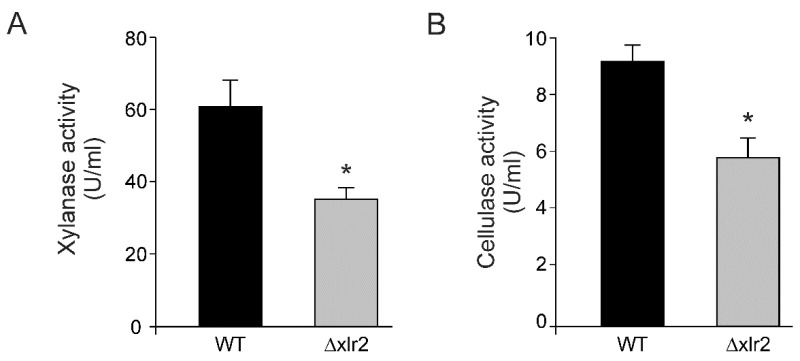
The enzymatic profile of *Trichoderma virens* Gv29.8 during submerged fermentation. (**A**) Xylanase activity using birch xylan as the carbon source and (**B**) cellulase activity using CMC as the carbon source. The results shown represent the mean value of three repetitions, and the error bars indicate the standard error of the mean. * Indicates that the Δxlr2 strain had significantly lower enzymatic activity than the WT (*p* < 0.05).

**Figure 4 ijms-25-05172-f004:**
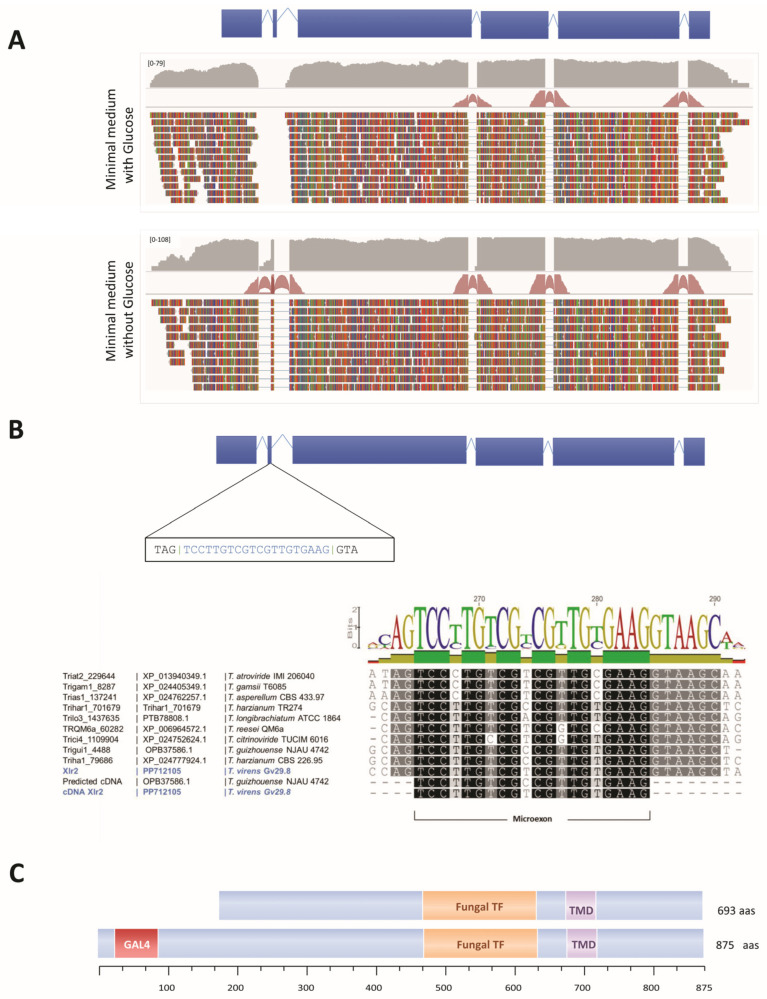
Gene model annotation of the Xlr2 (PP712105) gene from *Trichoderma virens* Gv29.8 containing two isoforms which differed by the inclusion/exclusion of exon 1 and the microexon. (**A**) Schematic of RNA-seq mapped to show the exons and transcripts differentially expressed in the presence or absence of glucose. The plot indicates different isoforms expressed in Vogel’s minimal medium with glucose or Vogel’s minimal medium without glucose. The mapping of the transcriptome reads from *T. virens* was visualised using the Integrative Genomics Viewer (IGV) visualization software (version 2.16.0) [48]. The schematic gene representation of Xlr2 from *T. virens* indicating the presence of a microexon expressed in medium without glucose and the alignment of the microexon of Xlr2 orthologs from other *Trichoderma* species is shown in blue blocks. (**B**) The percentage sequence identity of some Xlr2 homologs in *Trichoderma* spp. with more than 80% of the sequences in the alignment. The nucleotides identical to the Xlr2 amino acids of *T. virens* are indicated by a black background. (**C**) Scheme of the Xlr2 long and Xlr2 short gene of *T. virens* indicating the DNA-binding domains.

**Figure 5 ijms-25-05172-f005:**
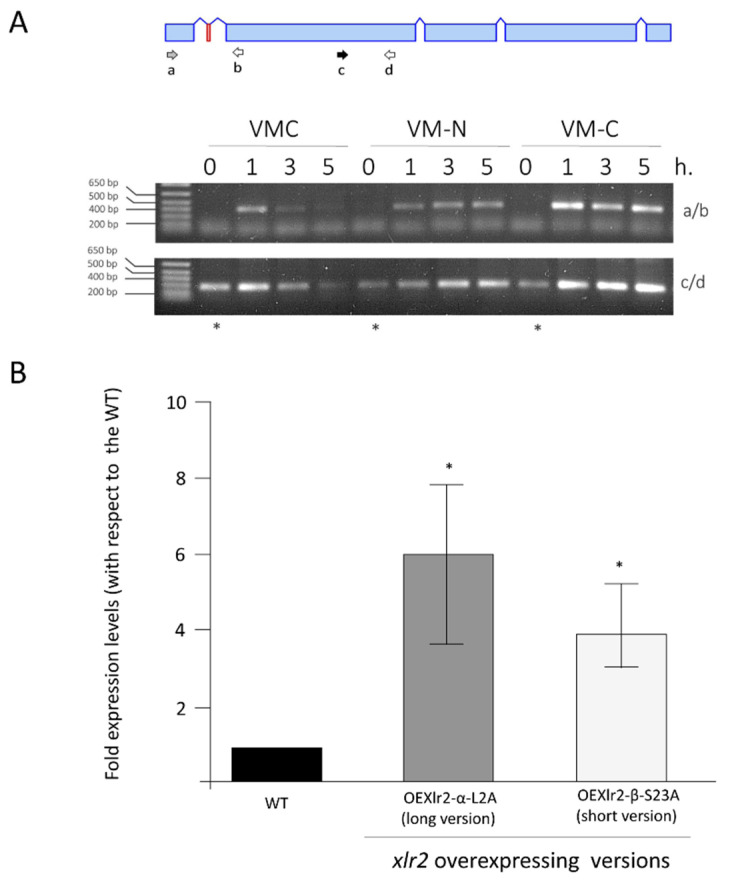
Overexpressing *Xlr2-α* and *Xlr2-β* in *Trichoderma virens* Gv29.8. (**A**) Schematic and RT-PCR of *Xlr2*: 0.8% agarose gel, cDNA synthesis of *Trichoderma* in the presence (VMC) or absence of glucose (VM-C) in Vogel’s minimal medium after 0, 1, 3, and 5 h of deprivation or not of a glucose (VM-C) or nitrogen (VM-N) source. The arrows represent the primers used to amplify the cDNA. (**B**) Relative quantification of the *Xlr2* gene expression and expression level of the overexpressing strains. The results shown represent the average value of three repetitions, and the error bars indicate the standard error of the mean. * Indicates that the OEXlr2-α-L2A or the OEXlr2-β-S23A strain had a significantly greater expression level than the WT (*p* < 0.05). The arrows “a” and “b” in A represent the primer localisation of AM-LU650 and RL-LU51LR, while arrows “c” and “d” represent the AM-LU708 and AM—LU821.

**Figure 6 ijms-25-05172-f006:**
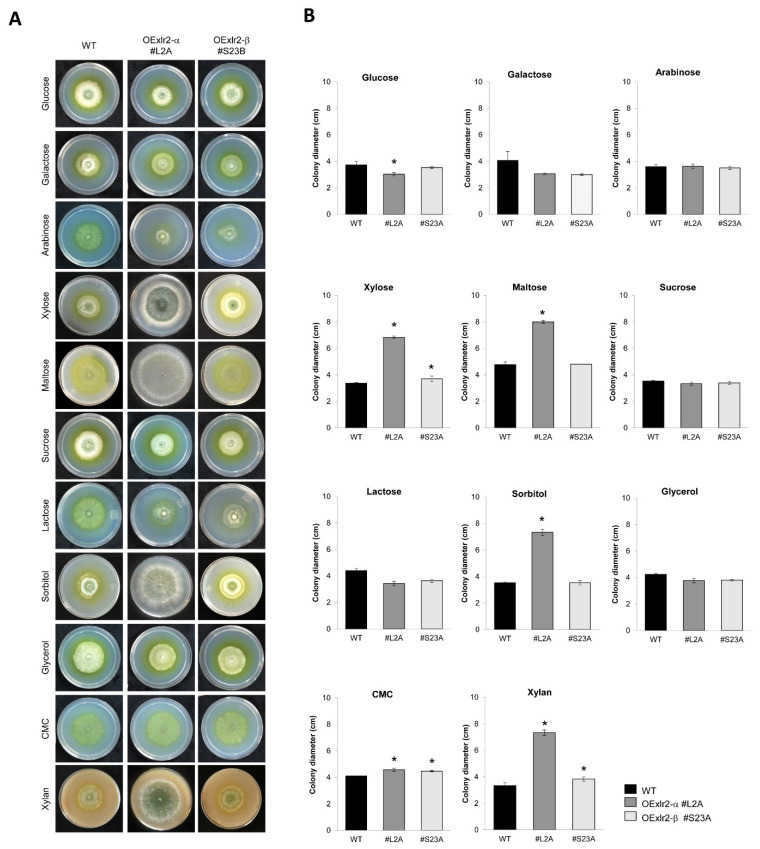
Growth profiles of the WT and overexpression (OEXlr2-α-L2A and OEXlr2-β-S23A) of *Trichoderma virens* Gv29.8 on different carbon sources. (**A**) Growth of the strains in Petri dishes for 3 d on Vogel’s medium supplemented with the indicated sugars at 25 °C. (**B**) Statistical analysis of the growth profile of the WT, OEXlr2-α-L2A, and OEXlr2-β-S23A strains of *T. virens*. The results shown represent the average value of three repetitions, and the error bars indicate the standard error of the mean. * Indicates that the OEXlr2-α-L2A or OEXlr2-β-S23A strain had significantly greater radial growth than the WT (*p* < 0.05).

**Figure 7 ijms-25-05172-f007:**
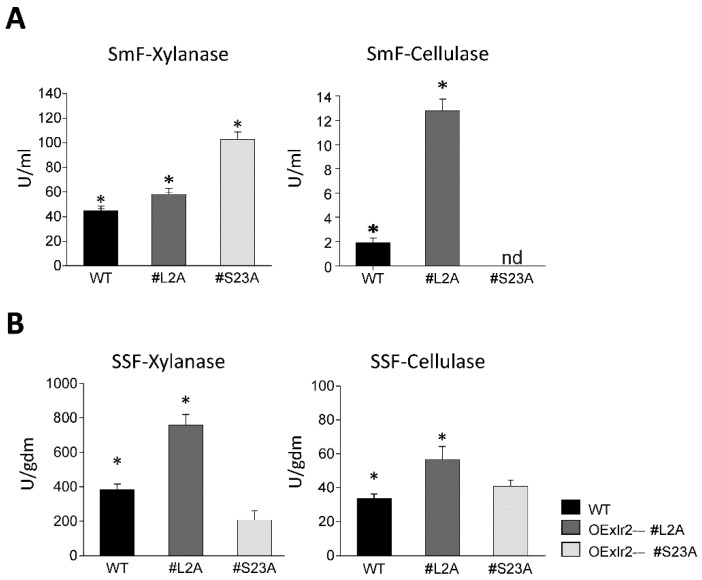
Evaluation of the xylanase and cellulase activity of *Trichoderma virens* Gv29.8 (WT) and strains overexpressing the *Xlr2* gene (OEXlr2-α-L2A (#L2A) and OEXlr2-β-S23A (#S23A)). The kinetics of enzymatic activity were determined during (**A**) submerged fermentation and (**B**) solid-state fermentation. The results shown represent the average value of three repetitions, and the error bars indicate the standard error of the mean. * Indicates that the OEXlr2-α-L2A or OEXlr2-β-S23A strain had significantly greater enzymatic activity than the WT (*p* < 0.05). nd means not activity detected in this condition.

## Data Availability

Data is contained within the article and Appendix A.

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
