# Peer review of "Unveiling a Microexon Switch: Novel Regulation of the Activities of Sugar Assimilation and Plant-Cell-Wall-Degrading Xylanases and Cellulases by Xlr2 in Trichoderma virens"

_ijms, 2024, doi:10.3390/ijms25105172_

Round 1

Reviewer 1 Report

Comments and Suggestions for Authors

The manuscript described the finding of a microexon in the Xlr2 gene of Trichoderma virens Gv29.8 from the RNA-Seq analysis data. The corrected long Xlr2a contained an extra N-terminal GAL4-like DNA-binding domain, while the short Xlr2b was designated as the original protein. An DXlr2 strain was used to study the growth effects under different carbon sources and cellulase and xylanase activities under liquid-state fermentation (LSF) with or without xylan, CMC, and glucose compared to the wild-type strain. Two overexpression strains, OEXlr2-α-L2A and OEXlr2-β-S23A, were constructed to study their effects on xylanase and cellulase activities. The results suggested that differential expression of the Xlr2 microexon is involved in controlling plant cell wall degrading enzymes. In summary, the current manuscript provided new insight into the regulatory role of an Xlr2a microexon under different carbon sources in controlling cellulase and xylanase activities, which was an important transcriptional regulation mechanism. However, the required information was lacking in the current manuscript. It is not recommended for the International Journal of Molecular Sciences but encourages resubmission after revision. The main concerns are listed below.

Major:

1.     The Trichoderma virens Gv29.8 Xlr2 gene was not submitted to GenBank to get an accession number, a basic publication rule.

2.     The genomes of several other T. virens were sequenced and could be found in the database. However, the microexon of the Xlr2 gene in other T. virens genomes was not analyzed. It is important to know if other T. virens Xyl2 genes showed the same characteristics.

3.     The information on the construction and verification of the DXlr2 strain was not provided. The Southern blotting data could be provided as supplementary materials.

4.     The bioinformatic tools were not described and cited in the Materials and Methods section.

5.     Some of the abbreviations, though common, should list the full names when first-time use, even shown in the figures. For example, ZnFD, fungal TF(D?), and TMD in Figure 1B.

6.     The fungal strain number and the accession number of the protein should be labeled in the figure and described in the legend. For example, those used for the alignment in Figure 1C.

7.     The genetic distance scale could be revealed in Figure 1D.

8.     Some of the gene and protein abbreviations were incorrectly used. For example, XYR1 or Xyr1? XlnR or Xlnr? AraR or araR? …… and many others.

9.     The protein's fungal strain name, number, and accession number should be described in the legend for Figure 4B. Some of the provided information in Figure 4B was incorrect.

10.  All the results of colony diameter shown in Figure 6B were labeled with a statistically significant mark *.

11.  Section 2.2. It was described that the TrpC (TrpC? or trpC?) promoter controlled the hygromycin phosphotransferase gene. However, it was described that trp1 (trp1?) promoter in Section 4.3 instead. Were there different types of vectors used?

12.  Some of the methods lacked references.

13.  The title was inappropriate because not all plant cell wall degrading enzymes were analyzed in this study.

14.  Abstract. Some of the descriptions were doubted. For example, it was stated that a short mRNA produced high glucose encodes a protein (Xlr2-b) lacking this DNA-binding domain. Why does a short mRNA produce high glucose?

15.  How can the authors confirm that the overexpressed Xlr2a proteins mediated the enhanced cellulase and xylanase activities but not the overexpressed Xlr2a transcripts?

Minor:

1.     The reference format didn’t follow the journal’s rules.

2.     The original gel images, which were additionally provided and represented in Figure 5, could be provided as supplementary materials.

3.     Back matter: Supplementary materials. Table S1 but not 1S, provided as the file name.

Comments on the Quality of English Language

Moderate editing of the English language is required.

Author Response

Major:

  1. The Trichoderma virens 8 Xlr2 gene was not submitted to GenBank to get an accession number, a basic publication rule.

Re: as requested by the reviewer, we submitted the Trichoderma virens Gv29.8 Xlr2 gene to GenBank: PP712105

  1. The genomes of several other virens were sequenced and could be found in the database. However, the microexon of the Xlr2 gene in other T. virens genomes was not analyzed. It is important to know if other T. virens Xyl2 genes showed the same characteristics.

Re: Thank you for your comment regarding the availability of other annotated genomes from Trichoderma virens. We appreciate your insight into the importance of analyzing multiple genomes from the same species to enhance the robustness of our study.

We identified only one additional annotated genome from the same species (T. virens FT-333). As suggested, we subsequently analyzed the presence of the microexon in this extra strain and is presented in Figures S3.

While we aimed to include multiple annotated genomes in our analysis, comprehensive annotations for publicly available T. virens genomes remain limited. Therefore, we acknowledge the constraint of analyzing only a single additional annotated genome in this study.

Nevertheless, our analysis of the microexon in the identified T. virens strain (FT-333), in conjunction with our primary study strain (Gv29.8), provides valuable insights into conserving this genomic feature within the species.

We greatly appreciate your feedback, and we will continue to explore avenues to incorporate additional annotated genomes into our analysis in future research endeavors. If you have any further suggestions or inquiries, please feel free to share them with us.

  1. The information on the construction and verification of the DXlr2 strain was not provided. The Southern blotting data could be provided as supplementary materials.

Re: Thanks so much for your recommendation, we included the Southern blot as well the confirmation of the absence of the open reading frame (orf) in our deletion mutants, This information is now incorporated in Figure 1S. 

  1. The bioinformatic tools were not described and cited in the Materials and Methods section.

Re: We described the bioinformatic methods in the text and included the corresponding citations.

  1. Some of the abbreviations, though common, should list the full names when first-time use, even shown in the figures. For example, ZnFD, fungal TF(D?), and TMD in Figure 1B.

Re: We have checked and corrected accordingly. The abbreviations in Figure 1B are described in the text and in the figure caption.

  1. The fungal strain number and the accession number of the protein should be labeled in the figure and described in the legend. For example, those used for the alignment in Figure 1C.

Re: We have updated Figure 1 following the reviewer’s comments.

  1. The genetic distance scale is revealed in Figure 1D.

Re: We have updated Figure 1D to include a genetic distance scale with additional fungal strains and protein numbers.

  1. Some of the gene and protein abbreviations were incorrectly used. For example, XYR1 or Xyr1? XlnR or Xlnr? AraR or araR? …… and many others.

Re: We have corrected the gene and protein abbreviations as suggested in the comments.

  1. The protein's fungal strain name, number, and accession number should be described in the legend for Figure 4B. Some of the provided information in Figure 4B was incorrect.

Re: Thanks so much for your suggestion; the information was incorporated in the figure rather than the figure legends.

  1. All the results of colony diameter shown in Figure 6B were labeled with a statistically significant mark *.

Re: We have updated the Figure 6B and labeled the * mark as (*) represents that OEXlr2-α-L2A or OEXlr2-β-S23A strains statistically significant increase in radial growth with a level of 95% confidence compared to the WT strain

  1. Section 2.2. It was described that the TrpC (TrpC? or trpC?) promoter controlled the hygromycin phosphotransferase gene. However, it was described that trp1 (trp1?) promoter in Section 4.3 instead. Were there different types of vectors used?

Re: We have checked and changed theTrp1 to TrpC.

  1. Some of the methods lacked references.

Re: We have checked and added two more citations in the method section.

  1. The title was inappropriate because not all plant cell wall degrading enzymes were analyzed in this study.

Re: We have updated our title to “Unveiling a Microexon Switch: Novel Regulation of Sugar Assimilation and Plant Cell Wall Degrading xylanases and cellulases Activities by Xlr2 in Trichoderma virens

  1. Some of the descriptions were doubted. For example, it was stated that a short mRNA produced high glucose encodes a protein (Xlr2-b) lacking this DNA-binding domain. Why does a short mRNA produce high glucose?

Re: Abstract was corrected accordingly

  1. How can the authors confirm that the overexpressed Xlr2a proteins mediated the enhanced cellulase and xylanase activities but not the overexpressed Xlr2a transcripts?

Re: We appreciate the insightful question raised by the reviewer regarding the specificity of the observed increase in cellulase and xylanase activities attributed to the overexpression of Xlr2α proteins versus transcripts alone. While our current findings strongly suggest a correlation between Xlr2α protein overexpression and enhanced enzymatic activities, we acknowledge that further experimentation is necessary to conclusively demonstrate causation.

As part of our ongoing research efforts, we plan to implement additional experiments to directly address this question. Specifically, we aim to quantify Xlr2α protein levels and correlate them with enzyme activities using protein quantification and activity assays. Additionally, control experiments isolating the effects of transcript overexpression will be conducted to distinguish between the roles of transcripts and proteins in mediating enzyme activity enhancement.

We recognize that this is an important aspect of our study and are committed to providing a thorough and well-supported answer. However, given the complexity of the experimental design and the need for rigorous validation, we anticipate that obtaining conclusive results may require additional time. We are dedicated to conducting these experiments diligently and will ensure that the results are reported accurately in future revisions of the manuscript. Your patience and understanding in this matter are greatly appreciated as we work towards addressing this valuable feedback.

Minor:

  1. The reference format didn’t follow the journal’s rules.

Re: We have updated the reference to follow the journal’s rules

  1. The original gel images, which were additionally provided and represented in Figure 5, could be provided as supplementary materials.

Re: We have included the original pictures and included as Figure S6.   

  1. Back matter: Supplementary materials. Table S1 but not 1S, provided as the file name.

Re: We have corrected to the Table S1.

Reviewer 2 Report

Comments and Suggestions for Authors

The authors of the manuscript ijms-2950429 describe functional microexon that regulate the expression of  Xylanase Regulator 2 Xlr2.

The manuscript is well-written and requires minor corrections before publication.

L65. I suggest underlining the fact that functional microexons have not yet been described.

L96. Use “biorefinery” instead of “bioenergy”. The biochemical route for lignocellulose biomass conversion, wherein Trichoderma Plant Cell Wall-Degrading Enzymes  (PCWDEs) are used, is not limited to bioenergy/biofuels.

L159-L173. It is necessary to clarify the Δxlr2 strain represented in Figure 2A and Figure S1. The authors mentioned, ”Three independent strains with identical phenotypes were selected.”  From these three strains, was only one selected?

L82-L197. Similar comment: It is necessary to clarify the strain used.

Sub-section 4.2. Submerged growth of the WT and Δxlr2  strains and microscopic analysis must be presented.

Author Response

The manuscript is well-written and requires minor corrections before publication.

  1. I suggest underlining the fact that functional microexons have not yet been described.

Re: We have added the information according to the Reviewer’s suggestion.

  1. Use “biorefinery” instead of “bioenergy”. The biochemical route for lignocellulose biomass conversion, wherein Trichoderma Plant Cell Wall-Degrading Enzymes (PCWDEs) are used, is not limited to bioenergy/biofuels.

Re: We have added the information according to the Reviewer’s suggestion.

  1. L159-L173. It is necessary to clarify the Δxlr2 strain represented in Figure 2A and Figure S1. The authors mentioned, “Three independent strains with identical phenotypes were selected.”  From these three strains, was only one selected?

Re: We used only strain Δxlr2 to compare with the wild type.

  1. L82-L197. Similar comment: It is necessary to clarify the strain used.

Re: We clarified in the text that we used only strain Δxlr2, which was deleted gene Xlr2.

  1. Sub-section 4.2. Submerged growth of the WT and Δxlr2 strains and microscopic analysis must be presented.

Re: We have updated Figure S2 for the supplementary of submerged growth of the WT and Δxlr2 strains and microscopic characteristics.

Round 2

Reviewer 1 Report

Comments and Suggestions for Authors

The manuscript described the finding of a microexon in the Xlr2 gene of Trichoderma virens Gv29.8 from the RNA-Seq analysis data. The corrected long Xlr2a contained an extra N-terminal GAL4-like DNA-binding domain, while the short Xlr2b was designated the original protein. AnXlr2 deletion strain was used to study the growth effects under different carbon sources and cellulase and xylanase activities under liquid-state fermentation (LSF) with or without xylan, CMC, and glucose compared to the wild-type strain. Two overexpression strains, OEXlr2-α-L2A and OEXlr2-β-S23A, were constructed to study their effects on xylanase and cellulase activities. The results suggested that differential expression of the Xlr2 microexon is involved in controlling plant cell wall degrading enzymes. It provided new insight into the regulatory role of an Xlr2a microexon under different carbon sources in controlling cellulase and xylanase activities, which was an important transcriptional regulation mechanism. The revised manuscript almost addressed all the questions and is recommended for the International Journal of Molecular Sciences after further modifications.

Major:

1.     The GenBank accession number of Trichoderma virens Gv29.8 Xlr2 gene, PP712105, could be described in the text and the figure legend.

2.     It would be better to have a bootstrap value of 1,000 to confirm the accuracy of the phylogenetic relationships for Figure 1D.

Minor:

1.     Zn(II)2Cys6 in the legend of Figure 1B.

2.     AraR but not araR in Figure 1C.

3.     The unit of the genetic distance scale can be described in the legend of Figure 1D. 

4.     It’s Xir2 and Xir2 cDNA in Figure 4B. The gene was shown in italics.

5.  It’s Dxir2 (xir2 mutant) in the legend of the back matter and that in supplementary materials Figure S2. 

Comments on the Quality of English Language

Moderate editing of the English language is required.

Author Response

Major:

  1. The GenBank accession number of Trichoderma virens Xlr2gene, PP712105, could be described in the text and the figure legend.

Thanks so much for your observation; the Xlr2 gene number was included in the text and the figure legend.

  1. It would be better to have a bootstrap value of 1,000 to confirm the accuracy of the phylogenetic relationships for Figure 1D.

The tree was generated again, and the Resampling method was set to Bootstrap with 1,000 replicates. 

Minor:

  1. Zn(II)2Cys6in the legend of Figure 1B.

This was changed

  1. AraR but not araR in Figure 1C.

This was fixed

  1. The unit of the genetic distance scale can be described in the legend of Figure 1D. 

The genetic distanced scale was described in the figure legend of Figure 1D.

  1. It’s Xir2and Xir2 cDNA in Figure 4B. The gene was shown in italics.

Thanks so much, we checked and the Xlr2 and Xlr2 cDNA is not in italics.

  1. It’s Dxir2 (xir2 mutant) in the legend of the back matter and that in supplementary materials Figure S2. 

Thanks so much. To avoid confusion, we removed the term (xlr2 mutant) in the figure legend of Figure 2S.

Moderate editing of the English language is required:

The final version of the current form of the manuscript was verified by one of the co-authors, an English language native speaker.